# Heavy Metals Bioindication Potential of the Common Weeds *Senecio vulgaris* L., *Polygonum aviculare* L. and *Poa annua* L.

**DOI:** 10.3390/molecules24152813

**Published:** 2019-08-01

**Authors:** Mirko Salinitro, Annalisa Tassoni, Sonia Casolari, Francesco de Laurentiis, Alessandro Zappi, Dora Melucci

**Affiliations:** 1Department of Biological Geological and Environmental Sciences, University of Bologna, Via Irnerio 42, 40126 Bologna, Italy; 2Department of Chemistry “G. Ciamician”, University of Bologna, Via Selmi 2, 40126 Bologna, Italy

**Keywords:** bioindication, heavy metals, urban soil, *Senecio vulgaris*, Poa annua, Polygonum aviculare, predictive models

## Abstract

In recent years, heavy metals (HMs) levels in soil and vegetation have increased considerably due to traffic pollution. These pollutants can be taken up from the soil through the root system. The ability of plants to accumulate HMs into their tissues can therefore be used to monitor soil pollution. The aim of this study was to test the ruderal species *Senecio vulgaris* L., *Polygonum aviculare* L., and *Poa annua* L., as possible candidates for biomonitoring Cu, Zn, Cd, Cr, Ni and Pb in multiple environments. The soils analyzed in this work came from three different environments (urban, woodland, and ultramafic), and therefore deeply differed for their metal content, texture, pH, and organic matter (OM) content. All urban soils were characterized by high OM content and presence of anthropogenic metals like Pb, Zn, Cd, and Cu. Woodland soils were sandy and characterized by low metal content and low OM content, and ultramafic soils had high Ni and Cr content. This soil variability affected the bioindication properties of the three studied species, leading to the exclusion of most metals (Zn, Cu, Cr, Cd, and Pb) and one species (*P. aviculare*) due to the lack of linear relations between metal in soil and metal in plants. *Senecio vulgaris* and *Poa annua*, conversely, appeared to be good indicators of Ni in all the soils tested. A high linear correlation between total Ni in soil and Ni concentration in *P. annua* shoots (*R*^2^ = 0.78) was found and similar results were achieved for *S. vulgaris* (*R*^2^ = 0.88).

## 1. Introduction

The increasing urbanization and industrialization in the last decades has resulted in spreading of heavy metals in the environment [1,2,3]. Since these elements are not degradable, they slowly accumulate in soil, becoming potentially hazardous to terrestrial and aquatic ecosystems and a risk to human and animal life [4,5]. Heavy metals (HMs) are naturally present in the environment as a result of either natural processes and human activities [6,7]. In natural ecosystems, HMs come from ultramafic rocks and ore minerals, and during weathering that leads to soil formation they can be released in the environment [8]. On the other hand, anthropogenic sources (i.e., vehicular traffic, mining activities, and refining processes) are nowadays the main responsible for HMs pollution [9,10]. Urban areas are recognized to be major sources for contaminants [11,12], and traffic is the primary source of HMs that accumulate in roadside soils [13] and street dust [14]. HMs are produced by vehicles exhaust emissions, as well as from the wear and tear of mechanical parts such as brakes, tires, and catalytic converters [13,15].

In recent years, it has been demonstrated that HMs levels in soil and vegetation have increased considerably because of traffic pollution, and the problem keeps growing with the increase of vehicular traffic [16]. This diffuse source of pollution in areas where people live and food is produced pose a serious threat to human health. In fact, these pollutants can enter plants directly via foliar absorption or be taken up from the soil through the root system [17] and undergo processes of biomagnification [18,19]. Despite the high toxicity of HMs for plants, when these elements are present in the soil at low concentrations, plants continue to grow healthy even while accumulating these metals. The ability of plants to accumulate HMs into their tissues can be used as monitoring tool to asses soil pollution [20], even though it has to be taken into account that many factors (i.e., climate, metal availability, etc.) could influence HMs uptake by plants.

Biomonitoring techniques using indicator plants (bioindication) are becoming common methods to detect toxic levels of HMs in the environments. Mosses and lichens, for example, are known to be the most sensitive indicators of atmospheric pollution [21,22], thus they are broadly used in urban environment. Unfortunately, because of the absence of roots and their restricted presence on hard substrate, they are not suitable for soil monitoring. Many authors agree that herbaceous plants could be effective biomonitoring tools, and some common species like the Dandelion, Nettle, and Broadleaf Plantain have already been successfully used [20,23,24,25]. Not all plants are suitable as indicators; some basic characteristics of good bioindicators have been listed by Witting [26]. An indicator plant should (i) accumulate one or several selected elements, (ii) have low sensitivity to the accumulated elements, (iii) have wide distribution in various environments, and (iv) show a correlation between metal accumulation and input into the ecosystem. 

Even when using the right plant, bioindication properties could be affected by other factors like soil properties complexation of HMs, oxidation state [8,27] and phenologic phase of the plants [28]. Seasonality plays an important role in determining HMs concentration in plant tissues. Despite some species grow all year long, it has been demonstrated that Cu, Fe, Mn, Pb, and Zn contents in Dandelion leaves collected in autumn were higher compared to those collected at the same sites in the spring [28]. Similar results were reported for alfalfa with regard to Mo content [29]. Soil properties, like cationic exchange capacity, clay content, pH, and organic matter content, are likely to change HMs availability to plants [30,31]. For example, Dai et al. [32] estimated that extractable Cd, Pb, and Zn levels in contaminated soils were positively correlated with organic matter contents. Moreover, low pH is optimal for metal availability since solubility has been shown to increase with decreasing pH [33,34]. Given the high variability of soils, bioindication cannot be considered a technique to precisely measure trace metals in soil, but rather a way to estimate them and their interaction with plants in some specific conditions. Therefore, it is of vital importance to assess indication properties of a species on several soils that differ for their HMs content and physical properties.

In this study we did not take into account the different forms of metals present in soils —e.g., Cr(III)/Cr(VI)— and metal complexes (e.g., Pb/tetraethyl-Pb) since we did not aim at the evaluation of the hazard of the different HMs and their forms. The aim of our study was to evaluate the ruderal species *Senecio vulgaris* L., *Polygonum aviculare* L., and *Poa annua* L. as possible candidates for biomonitoring Cu, Zn, Cd, Cr, Ni, and Pb in multiple environments, with special attention to the urban ones. These species have never been tested for biomonitoring potential, but since their wide diffusion in all anthropic environment, they could help in detailed biomonitoring campaigns in urban areas. Furthermore, we aimed to assess how different types of soils can affect the predictive potential of these species.

## 2. Materials and Methods

### 2.1. Samples Collection

For this study, three common weed species that have the basic characteristics of “good bioindicators” according to Witting [26] were selected. All three species are ruderal plants, which makes them common in all anthropic habitats.
(1)*Senecio vulgaris* L. (groundsel) is an annual plant of the Asteraceae family. Originally having an Eurasiatic distribution, today it has become subcosmopolitan worldwide. The species is common everywhere, and it grows prolifically in disturbed habitats like road margins, arable fields, and gardens. It preferably grows on clayey soil rich in nitrogen and organic matter.(2)*Poa annua* L. (annual bluegrass) is an annual plant of the Poaceae family. Originally having an Eurasiatic distribution, today it is widely naturalized in the temperate areas of the globe. It is a pioneer species that grows in trampled areas, gardens, and roads margins, in nitrogen-rich soils.(3)*Polygonum aviculare* L. (common knotgrass) is an annual plant of the Polygonaceae family. The species is cosmopolitan, and because of its high variability it is adaptable to several habitats. It grows on all soil types and it is resistant to trampling. It is widespread in urban areas, arable fields, but also woodland margins.

To maximize the metal content in plant tissues, plants were harvested close to the end of their life cycle, therefore during the fruiting season (April–May 2017 for *P. annua* and *S. vulgaris*; October 2017 for *P. aviculare*). Only the aerial parts of the plants were taken. After collection, plants where thoroughly washed with deionized water, then oven dried at 50 °C until constant weight. Dried samples were powdered with an Ultraturrax A11 basic Analytical mill (IKA^®^, Staufen, Germany), then stored at room temperature until analysis.

In order to have high heterogeneity of soil conditions, the three herbaceous species were harvested in eight stations belonging to three different habitats (Figure 1). Five stations were from urban environment (BO7, BO8, MI3, MI4, and MI9; “BO” indicating samples collected in Bologna and “MI” collected in Milan), two stations were from woodland environment (NAT1 and NAT5; “NAT” indicating natural environment) and one was an ultramafic station from Mount Prinzera (NAT8). The choice criteria were: most and least polluted stations in urban environments, random choice among woodland station (since they were all similar), and the only ultramafic station.

In every station, the three species were all present simultaneously, growing in the same bulk soil. The soil was sampled at a depth of 0–10 cm exactly below the plants, and at least 5 soil subsamples were collected in each location. The subsamples were then mixed together to form one bulk sample (of ~2 kg). The bulk soil samples were homogenized and sieved at 0.5 cm to exclude stones and other coarse particles, then oven dried at 50 °C until constant weight. Dried soil samples were further sieved at 0.1 cm, then stored at room temperature until analysis.

### 2.2. Soil Digestion

All chemicals used were suprapure and were purchased from Sigma-Aldrich (Saint Louis, MO, USA): 69% (*w*/*w*) HNO_3_, HCl 37% (*w*/*w*), 35% (*w*/*w*) H_2_O_2_, 96% (*w*/*w*) sulfuric acid, ammonium citrate, iron (II) ammonium sulfate hexahydrate, and potassium dichromate. RC syringes filter (Fisher Scientific, Hampton, NH, USA) (porosity 0.45 µm; diameter 22 mm) were employed to filter solutions after metals extraction.

To perform soil digestion, a modified a version of the US EPA 3050b method was used. The dried and sieved soil (0.1 cm) was finely grinded in a mortar; then, approximately 1 g of soil and 5 mL of 69% (*w*/*w*) HNO_3_ were put in a Pyrex 100-mL calibrated test tube. The tube was connected to a Vigreux column, and was then placed in a special housing on a heating plate at 150 °C. The system was left in reflux mode for 30 min. Subsequently, the tube was cooled in an ice-bath, then 5 mL of 35% (*w*/*w*) H_2_O_2_ was added, and the addition of H_2_O_2_ drops continued until the solution in the tube stopped boiling. Then, 10 mL of 37% (*w*/*w*) HCl were introduced, and another reflux step was applied for 15 min. Once digestion was completed, the solution was cooled down and filtered. Finally, the liquid phase was transferred into a 50-mL flask and brought to the final volume with 0.5 M HNO_3_. The total concentration of metals in soils (Total Metal, TM) was measured in µg g^−1^ (briefly indicated as “ppm”). Blank digestions (without soil) were carried out using the same reagents as described above.

### 2.3. Plant Digestion

Acid digestion on plant shoots was carried using a modified protocol adapted from Huang et al. [35]. An aliquot of plant powdered shoots, between 0.05 and 0.1 g, was placed in a 10 mL glass tube and 2 mL of 69% (*w*/*w*) HNO_3_ were added. A pre-digestion phase was obtained by leaving the tubes at room temperature for 24 h. Then, the tubes were placed on a hot plate at 75 °C for 1 h, and subsequently the temperature was increased to 125 °C for another 1 h. During the digestion the tubes were left open without any reflux system.

In the 2 h digestion, the volume of acid reduced to about 1 mL, and then it was transferred in a 10 mL flask and brought to the final volume with Milli-Q grade water (Millipore, Bedford, MA, USA) to obtain a digestate with about 6–7% (*w*/*w*) HNO_3_. Generally, no plant residues were visible, but for the sake of sureness, filter syringes were used. The concentration of metals in plants (Plant Metal, PM) was measured in µg g^−1^ (briefly indicated as “ppm”). Blank digestions (without plants) were carried out using the same reagents as described above.

### 2.4. Extraction of Bioavailable Metal Fraction from Soil

The dried and sieved soil (0.1 cm) was finely grinded in a mortar, then sieved with a 0.5 mm mesh. Five grams of sieved soil was then transferred to an extraction bottle in which 5 mL of 2% (w/v) ammonium citrate solution was added. The obtained mixture was shaken on an end-over-end tube roller mixer at 30 rpm for 1 h at 20 °C. The extracts were immediately separated by decantation for few minutes, followed by centrifugation for 10 min at about 3000 Ug. The supernatant was recovered and the liquid was stored in a polyethylene container at 4 °C until analysis. The concentration of bioavailable metals in soil (BM) was measured in µg g^−1^ (briefly indicated as “ppm”). Blank extractions (without soil) was carried out using the same reagents as described above [36].

### 2.5. Determination of Metal Concentration in Soils and Plants

The concentration of metal in soils and plants was quantified through Atomic Absorption Spectroscopy (AAS). AAS measurements were performed using an Atomic Absorption Spectrometer AAnalyst 400 (Perkin-Elmer, Waltham, MA, USA), equipped with a deuterium background corrector, Autosampler AS-72 (Perkin-Elmer) and a HGA 800 graphite furnace (Perkin-Elmer). Single-element Lumina (Perkin-Elmer) hollow-cathode lamps were used. All measurements were carried out using default program for ashing and atomization curves for each element, at the detailed instrumental conditions are reported in Table 1. All the elements, except Zinc, were determined by electro-thermal AAS (ET-AAS), employing argon at flow rate 250 mL min^−1^ in all steps except during atomization (0 mL min^−1^).

Zinc was analyzed by flame atomic absorption (FAAS) employing Acetylene (4.10 L min^−1^) and air (10 L min^−1^). 

For each of the six analyzed metals (Cu, Pb, Cd, Cr, Ni, and Zn) a calibration line was created. Standards for calibration lines were purchased by Merck (Darmstadt, Germany). Three standard solutions were prepared for each metal and “outer” standard concentrations were selected in order to stay in the linear range of each analyte, as tabulated in the software WinLab 32 (Perkin-Elmer). Peak area was used as the analytical signal, after verifying that peak height never overcame 0.6 AU, in order to stay in the absorbance linear range. Before each analysis, a blank sample was analyzed and the peak area of the sample was subtracted to the previous blank one. For each calibration line, the limit of detection (LoD) was computed by applying the equation LoD = (*K s_y/x_*)/*b* [37], where *s_y/x_* and *b* are the estimated regression standard deviation and the slope of the relevant analytical calibration function, respectively. *K* = 3 was chosen in order to obtain the limits of detection. It was verified that LoD never overcame the lowest standard concentration. When analyses had to be carried out in several days, every day three standards were analyzed and projected on the calibration line, to verify its validity. Three replicates were analyzed for each standard and sample. The injected volume was 20 μL for each analysis. Samples were properly diluted in order to obtain a signal in the calibration range, and the dilution factor was kept into account to calculate the metal concentration in the sample.

In order to analyze Cd, Cr, Ni, and Pb by AAS, some matrix modifiers were necessary. In particular, Mg(NO_3_)_2_ (Perkin-Elmer) for Cd and Cr, PdCl_2_ (Fluka, Honeywell, Morris Planes, NJ, USA) for Cd, and NH_4_H_2_PO_4_ (Sigma Aldrich) for Pb. A solution (20 μL mL^−1^) containing all of the modifiers was added to each sample; final concentrations: 200 mg L^−1^ for Mg(NO_3_)_2_ and 2.3 mg L^−1^ for PdCl_2_, 4 mg L^−1^ for NH_4_H_2_PO_4_. It was also verified that the presence of an unnecessary modifiers did not influence the measurements of other metals (as Cu and Zn, which did not require any modifier).

### 2.6. Determination of Organic Matter and Granulometry of Soil

The percentage of organic matter in soils (OM) was determined by two experimental methods: titration and Loss on Ignition. The percentage of inorganic matter (IM) was calculated as 100%-OM.

To measure OM, the titration was carried out following the method in Walkley [38]. Half a gram of soil, 10 mL of potassium dichromate 0.167 M, and 20 mL of 96% (*w*/*w*) sulfuric acid were placed in a 500-mL conical flask, slowly percolating along the internal walls of the flask, not to overheat the mixture. The flask was covered with watch glass and left to rest for 30 min. Then the reaction was interrupted by adding 200 mL of distilled water, previously cooled in the refrigerator. A few drops of ferroin (redox indicator) were added, and titration was carried out with a solution of iron(II) ammonium sulfate hexahydrate 0.5 M until the color changed. At the same time, a blank test was performed with 10 mL of dichromate, 20 mL of sulfuric acid, and 200 mL of distilled water. The following expression was used for the calculation of organic carbon (C) expressed in g kg^−1^.
(1)$$C=3.9\cdot \frac{\left(B-A\right)}{MSoil}\cdot \mathrm{MFe}$$
where *B* = volume of the solution of iron (II) ammonium sulfate hexahydrate used in the titration of the blank test, expressed in mL; *A* = volume of the solution of iron (II) ammonium sulfate hexahydrate used in the titration of the sample solution, expressed in mL; MFe = effective molarity of the solution of iron (II) ammonium sulfate hexahydrate; and MSoil = mass of the soil sample, expressed in grams. To transform g kg^−1^ of organic carbon into the corresponding organic substance content, a conversion factor is applied: % OM_titr = %OM_titr∙1.724. 

To validate OM content found by titration, we compared the results with the one found by Loss on Ignition method (official method for the determination of OM) as explained in Storer [39]. The two methods were comparable and gave similar results, therefore the titration method results were validated and used for statistical elaborations. 

Granulometry of the samples was assessed by sieving samples with gradually smaller mashes and weighing the fraction held into each mesh. Four classes of granulometry were defined: particles >0.5 mm (coarse), particles between 0.5 and 0.25 mm (medium), particles between 0.25 mm and 63 µm (fine), and particles <63 µm (ultrafine).

### 2.7. Data Analysis

The matrix containing all collected data was composed by 86 observation (objects) and 43 variables.

We used chemometrics to extract useful information from our dataset and in particular to create and validate models. Chemometrics was applied both in univariate mode (analysis of correlation and creation of linear regression) and in multivariate mode (Principal Components Analysis (PCA)) [39,40]. To create linear models, the Multiple Linear Regression tool (MLR) was applied. In order to validate MLR models, besides considering the model *p*-values (ANOVA test), which should be close to the null value, the same data used to create the model were projected onto it, both in “calibration” mode (blue dots in response plots) and by leave-one-out cross-validation (LOO-CV) (red dots in response plots). Projection in calibration mode means that once the model is created, data are projected onto it as they are. LOO-CV, on the other side, creates as many models as the number of samples, leaving each time one sample out from the model creation and projecting it onto such model. In this way, each sample is treated as if it was an external data used to validate the regression performance of the overall model. Both in calibration and in LOO-CV, response values of each sample are recalculated by projection. Then, two further response lines are computed (blue for calibration mode; red for LOO-CV), in which the independent variables are the known (experimental) response and the dependent variables are the recalculated values that appear in the response plot. The predictive model performances are evaluated by the parameters of these lines. In a perfect case, the recalculated responses would be exactly equal to the known ones, thus the response lines would be the bisector of the response plot, with slope = 1, offset = 0, and R^2^ = 1. Model performances are considered acceptable if the response line parameters are close to these ideal values. A further parameter of the response lines is root mean squared error (RMSE), which is a sort of sum of distances between the known responses and the recalculated ones, therefore it should be close to zero.

The potential suitability of a species as bioindicator of one or more metals was validated in two steps. Firstly, we tested the correlation between bioaccumulation factor (BAF) calculated on TM and BAF calculated on BM, if that correlation was good (between 0.7 and 1), we tested the correlation between metal content in plants (PM) and metal content in soil (TM). In Table 2, the empirical rules adopted to evaluate correlation “goodness” between variables are shown. Only plants showing “high” or “excellent” correlation values for both validation steps were used in the creation of predictive models. All statistical analyses and graphical elaborations were performed using the software The Unscrambler 10.4 (CAMO Analytics, Oslo, Norway).

## 3. Results

### 3.1. Heavy Metals in Soil

The eight soils analyzed appeared to be strongly different one the other and characterized by various metal content, texture, pH, and OM content. Soil variability was explored trough a PCA (Figure 2) that showed a clustering of soils according to the area of collection.

Heavy metals (HMs) concentrations, were closely linked to the levels of anthropogenic activity for urban and woodland soils, while were mainly from geogenic origin in ultramafic soils. The analyzed metals were present in the decreasing order: Zn > Cu > Cr > Ni > Cd for urban areas, Zn > Cu > Ni > Cr > Cd for woodland areas and Ni > Cr > Zn > Cu > Pb > Cd ultramafic areas. All urban soils were characterized by medium to fine granulometry, high OM content especially for Milan samples and presence of anthropogenic metals like Pb, Zn, Cd, and Cu. Woodland soils from Bologna were quite similar to urban ones with a slightly coarser texture and lower pH levels. Milan woodland soils were sandy and characterized by low metal content and low OM content. Finally, Prinzera soils, because of their ultramafic origin, had high Ni and Cr content; moreover, they are characterized by coarser granulometry (Figure 3). Metal concentrations (both total and bioavailable) in each soil are summarized in Table 3. The most polluted soils were MI3 and MI4, while the least polluted were NAT1 and NAT5, except for Ni and Cr where NA8 had the highest values.

From Table 3, it can be noted that Zn content is one order of magnitude higher in urban soils from Milan (>1000 ppm) if compared with other soils, while the lowest values can be found in woodland soils (about 100 ppm). The situation was similar for copper: urban soils from Milan had the highest values (>500 ppm) and woodland soils had lowest levels of Cu (~10–50 ppm). Again, for Pb, a similar situation to the previous two metals was detectable: the concentrations varied from approximately 100 to 500 ppm for Milan urban soils to approximately 10 to 20 ppm for woodland soils. These first three metals were therefore connected to anthropogenic activities. The situation of Cd is substantially different, which had similar levels (~0.35 ppm) in all the analyzed soils. Finally, Cr and Ni showed a wide range of concentrations in soils, independently of the origin of the soil. Nevertheless, the highest concentrations of these metals were found in the ultramafic soils from Prinzera (Ni: ~1800 ppm; Cr: ~500 ppm).

### 3.2. Species and Metal Selection

A preliminary exploration tested the correlation between the BAF calculated on TM (BAF_TM) and BAF calculated on BM (BAF_BM). This step was useful to evaluate if the response of our species was consistent both in considering the total metal in the soil and the bioavailable fraction. Only plants and metals that had high correlation values were kept as candidates for bioindication. Species with high correlation values for certain metals were likely to give consistent information on total and bioavailable metals in soil simultaneously.

High correlation values (Table 4) between BAF_TM and BAF_BM were found for all species in at least two metals each. From this preliminary screening, *S. vulgaris* appeared to be a possible candidate for bioindication of Cu, Pb, Cd, and Ni. *P. aviculare* was found to be a potential bioindicator of Pb, Cr, Cd, and Ni. Finally, *P. annua* could be a possible bioindicator for Pb and Ni. 

Metal concentrations for all plants are reported in Table 5.

These data were used to carry out another explorative analysis focused on the correlation between plant metal content (PM) and the soil metal content (TM). High correlations were found for two species, but only in the case of Ni (Table 6); therefore, the potential of *P. annua* and *S. vulgaris* as bioindicators for Ni was further explored and modeled.

Some other relevant correlations were found in *P. aviculare* for Cd and Pb and for *P. annua* for Cr, but these species did not show any linear relation with the metal when furtherly tested. Hence, attention is focused on Ni in the following section, with the aim of creating predictive models for Ni bioindication using *P. annua* and *S. vulgaris*.

### 3.3. Bioindication of Ni Using P. annua

The strong linear relation (*R*^2^ = 0.841) between total Ni in soil and Ni concentration in *P. annua* shoots is shown in the table enclosed in Figure 3A. This linear relation made it possible the creation of a MLR model with predictive potential (response plot in Figure 3A). The performance of the models was considered relevant to bioindication purposes since this model appeared reliable when tested by ANOVA (*p*-value related to the *F* parameter <0.05). Both in calibration (blue dots) and cross-validation (red dots), the two lines and the values almost overlapped. Except for high Ni values, the model appeared very accurate in the prevision of total Ni in soil using as input data Ni concentration measured in *P. annua* plants.

An even better relation (*R*^2^ = 0.928) was obtained when considering the bioavailable fraction of Ni compared to Ni content in *P. annua* shoots (table enclosed in Figure 3B). The connected MLR model showed an even higher predictive potential, with high accuracy for the whole range of Ni values (response plot in Figure 3B). The high *R* values and the solidity of both models confirmed that *P. annua* can be used as reliable Ni bioindicator.

### 3.4. Bioindication of Ni Using S. vulgaris 

The situation of *S. vulgaris* is similar to the one of *P. annua.* The data showed a clear linear relation (*R*^2^ = 0.908) between Ni in soil and Ni in plant (table enclosed in Figure 4A). From this strong relation, the creation of a predictive MLR model was also possible (response plot in Figure 4A). For low Ni values the performance of the models was high both in calibration and cross-validation mode. While for high Ni values, results obtained in cross-validation were slightly different from the calibrations ones. The model appeared reliable when tested by ANOVA (*p*-value related to the *F* parameter <0.05) therefore significant for bioindication purposes.

An even better linear relation (*R*^2^ = 0.969) was obtained when considering the bioavailable Ni pool in soil compared to *P. annua* Ni content (table enclosed in Figure 4B). The connected MLR model showed therefore an even higher predictive potential (response plot in Figure 4B). In this case, for all Ni concentrations the performance of the model was similar, with the data forecasted in cross-validation almost overlapped to the ones calculated in calibration. This high similarity of the two sets gave strength to the model, which can be considered highly reliable for available Ni prevision in soil.

## 4. Discussion

The analyzed soils in this work were selected from three different environments characterized by different levels and sources of HMs pollution. Urban soils from Milan and Bologna, despite some peculiarities connected to parent materials that contributed to their pedogenesis, were characterized by similar amounts and types of HMs. All urban surfaces, in fact, receive deposits that mainly come from anthropic activities, like vehicle emissions, industrial discharges, domestic heating, and material weathering [41,42].

Street dust and top roadside soils in urban areas are typical sinks of HMs from atmospheric deposition and water runoff. Key HMs in these zones are Pb from gasoline additives; Cu, Zn, and Cd from car components; tire abrasion; lubricants; and industrial emissions [43,44]. MI3 and MI4 (Table 4) had the most polluted soils as they were collected at a very busy street crossing. This relation underlined the important role of vehicular traffic in contributing to soil pollution.

Woodland soils were collected in natural areas that were not influenced by anthropic activities; trace elements detected in such soils were the one originally present in parent materials. However, a small contribution form diffuse sources of HMs pollution, like vehicles emission, cannot be excluded for these samples. It has been reported that fine particulate for example coming from tires and brakes abrasion may fall far away from source location [45]. Finally, ultramafic soil, despite being collected far from anthropic source of pollution, is still present at physiologic high levels of Ni and Cr. The level of anthropic HMs (Zn, Cu, Pb, Cd) was low, demonstrating the naturalness of this environment, but the ultramafic rock on which pedogenesis took place was highly enriched in Ni and Cr, now found in soils [46].

The marked heterogeneity of sampled soils was the main obstacle to the creation plant-soil linear relations in the absorption of metals. The investigated soils showed a wide range of properties (like different texture, pH, OM content, etc.) that deeply affected the bioavailability of metal to plants [47,48].

Moreover, it is widely known that trace elements are not found in plant tissues with the same the proportion of their concentrations in soil [49]. The uptake of these elements by plants is selective: essential nutrients (like Zn, Cu, and Mn) are actively taken and show a more linear relation to soil concentration if compared to nonessential nutrient [47,50]. This active absorption of micronutrients eventually results in a greater translocations and concentration in plant shoots, but also to a higher toxicity if compared to nonessential nutrient [51]. 

In agreement with these uptake mechanisms, the species investigated in this paper showed higher correlations for Zn and Ni (micronutrients) than for nonessential ones (Table 6). Despite the correlation coefficient for Zn of about 0.6 for all species (Table. 6), no linear relations were found for this metal. On the other hand, *P. annua* (*R*^2^ = 0.87) and *S. vulgaris* (*R*^2^ = 0.73) showed a linear relation with Ni in soil, as similarly found for *Taraxacum officinale* [25]. 

Another possible reason for the lack of linear relations between metal content in soils and plants was probably due to the poor translocation of these elements from root to shoot. Other studies in fact demonstrated that when nonessential metals are present at high concentration in soil, most herbaceous plants tend to use exclusion strategies to prevent the uptake of these toxic elements [52,53]. This phenomenon was observed for Cd in *Halophyla ovalis* [51] and for Pb [54]. Species unable to prevent root absorption, instead limit the translocation to shoots keeping the majority of toxic elements stored in roots [55]. 

In the present study roots were not collected, so data about metal concentration in belowground organs were not available, but an extensive literature demonstrated that root metal concentration better correlates with soil HM concentration if compared to shoots [56,57]. However, shoots are the most used parts for bioindication purposes, due to their visibility and easiness of collection. For this reason, the use of plants with limited translocation of HMs in aboveground parts has low practical application [56]. 

The bioaccumulation factor (BAF), a parameter that quantifies the element transfer from soil to plant, was found to be lower than one for almost all samples in reference to all metals. Due to this low BAF, all the studied species can be considered nonaccumulators, according to van der Ent et al. [58] metal accumulator plants must have this parameter always higher than one. Interestingly, several BAF values were found to be above one for Cd (Table 5), and similar results were also reported for *P. major* by Galal & Shehata [23], demonstrating that this metal at low concentrations can be easily uptaken and transferred to aerial parts. 

*P. annua* and *S. vulgaris* (Figure 5) had similar BAF values for Ni uptake in all soils, making them suitable for bioindication purposes.

This is in line with the guidelines from the EPA [59], in which it is stated that good indicator plants should keep this parameter constant in several soil conditions. Moreover, Ni is very mobile inside the plant, and is transported (binded by organic acids) through the xylematic flow from roots to shoots [60]. This work demonstrated that aerial parts of the common weeds *P. annua* and *S. vulgaris* can be used as environmental indicators of Ni pollution in soil. Similar results were achieved for *Urtica dioica, Taraxacum officinale, Plantago major*, and two Trifolium species for Pb, Mn, and Cu [20]. The use of common weeds can be a valid alternative to the use of lichens in assessing HMs levels in cities, especially because these herbaceous plants are common and easily recognizable. 

Interestingly, Malizia et al. [20] achieved similar results when assessing HMs in soil using lichens or using herbaceous plants for Cu, Zn and Pb, in the city of Rome. This promising result should encourage the research on common weeds as valid alternative to “lichens biomonitoring” in urban areas.

Despite the extensive literature about biomonitoring using herbaceous species [20,61], most studies focused on *Taraxacum officinale* while only few took into account other species [20]. 

This study results highlighted the possibility to find new species suitable for bioindication of metal pollution in anthropic environment. The importance of having several bioindicator species in each environment has been underlined by Phillips et al. [57] who suggested a “multi-species” approach to bioindication, in order to obtain more precise results. Finally, we demonstrated the possibility to create predictive models when strong linear relations are present between soil Ni and plant Ni. This chemometric approach was not aimed at the replacement of collection and analysis of samples, but, instead, to give support at the results achieved with traditional field samplings and lab analysis. 

## 5. Conclusions

Our results about the possible use of *S. vulgaris, P. aviculare*, and *P. annua,* as HMs bioindicators, showed that metal concentrations in soils and plants mostly do not correlate under natural growth conditions. Despite metals found in soils being present in plants, the concentration in aboveground organs is deeply influenced by soil properties and plant translocation. For Zn, Cu, Pb, Cr, and Cd, none of the studied species was enough equipped to be a good bioindicator; moreover, *P. aviculare* was found to be inappropriate even as Ni indicator. Nevertheless, the present work demonstrated the feasibility to use *P. annua* and *S. vulgaris* as bioindicators of Ni pollution in soil. The two species were reliable indicators of total and bioavailable Ni fraction. The good results achieved with this two species allowed us to develop models based on our data, which were able to forecast with great accuracy Ni concentration in soil from Ni in plants. However, these promising results need a definitive validation using a greater number of samples.

## Figures and Tables

**Figure 1 molecules-24-02813-f001:**
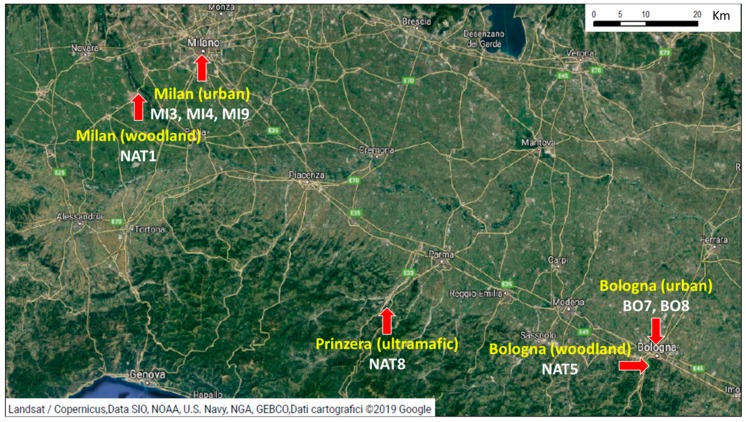
Sampling locations of soils and plants used in the study. In each station one soil sample and three plant species have been collected.

**Figure 2 molecules-24-02813-f002:**
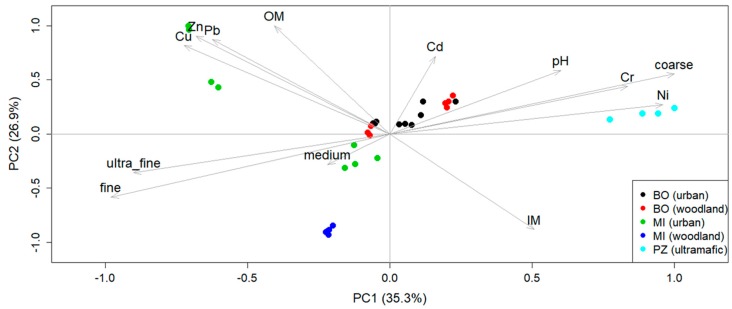
Soil clustering after principal component analysis (PCA). The input data were the soil variables of granulometry, OM, IM, and total heavy metals concentration.

**Figure 3 molecules-24-02813-f003:**
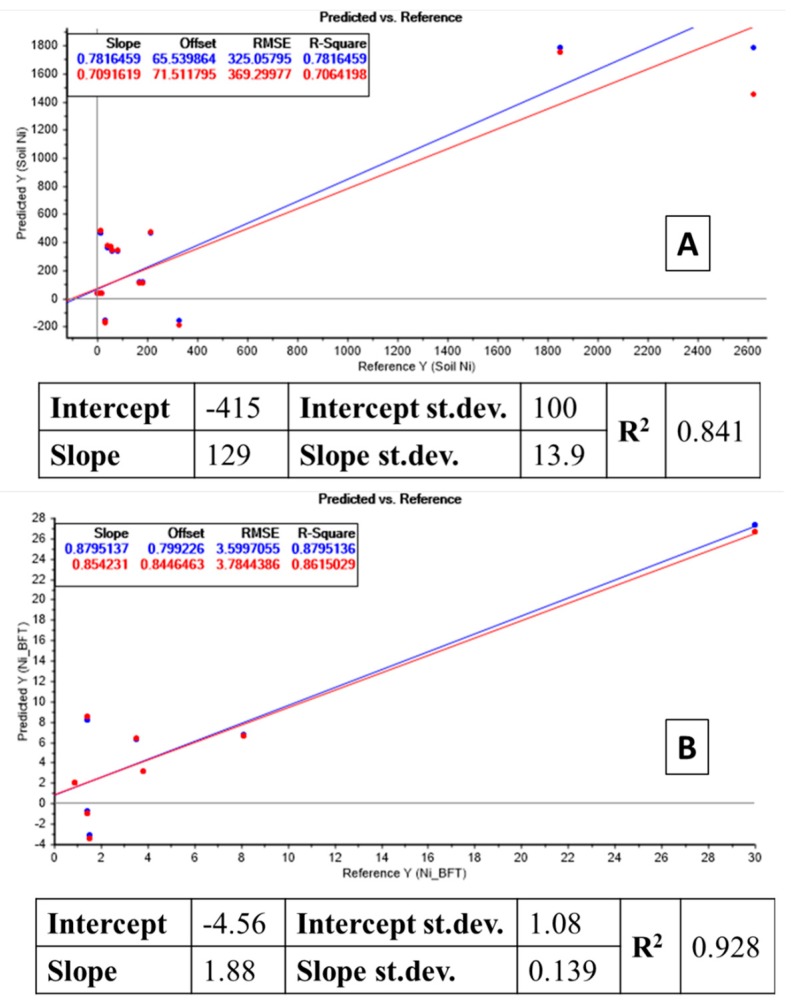
(**A**) Table: Linear regression between total Ni concentration in soil and Ni in *P. annua* shoots. Plot: recalculated total soil Ni by the model, input data derived from the linear relation between total Ni in soil, and Ni in plant. Blue dots: forecasted soil Ni concentrations in calibration mode (all soil data were used as input). Red dots: forecasted soil Ni concentrations in cross-validation mode, excluding one soil data at a time (leave-one-out mode). (**B**) Table 2: Linear regression between bioavailable Ni concentration in soil and Ni in *P. annua* shoots. Plot: Recalculated total soil Ni by the model; input data are derived from the linear relation between total Ni in soil and Ni in plant. Blue dots: forecasted soil Ni concentrations in calibration mode (all soil data were used as input). Red dots: forecasted soil Ni concentrations in cross-validation mode, excluding one soil data at a time (leave-one-out mode).

**Figure 4 molecules-24-02813-f004:**
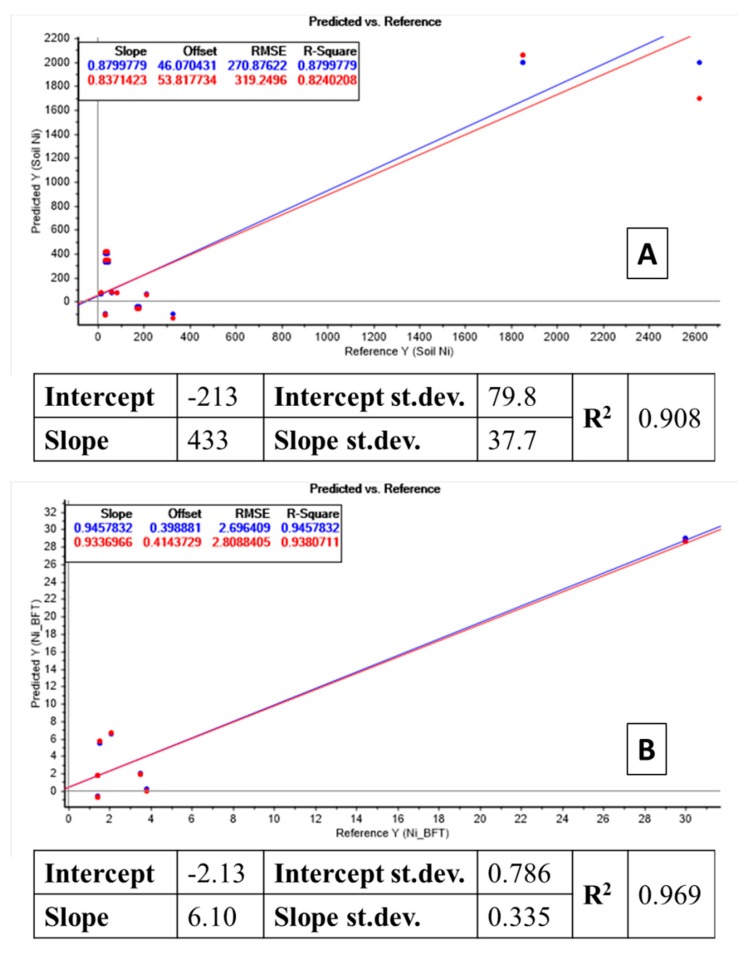
(**A**) Table: Linear regression between total Ni concentration in soil and Ni in *S. vulgaris* shoots. Plot: recalculated total soil Ni by the model, input data derived from the linear relation between total Ni in soil and Ni in plant. Blue dots: forecasted soil Ni concentrations in calibration mode (all soil data were used as input). Red dots: forecasted soil Ni concentrations in cross-validation mode, excluding one soil data at a time (leave-one-out mode). (**B**) Table: Linear regression between bioavailable Ni concentration in soil and Ni in *S. vulgaris* shoots. Plot: Recalculated total soil Ni by the model; input data are derived from the linear relation between total Ni in soil and Ni in plant. Blue dots: forecasted soil Ni concentrations in calibration mode (all soil data were used as input). Red dots: forecasted soil Ni concentrations in cross-validation mode, excluding one soil data at a time (leave-one-out mode).

**Figure 5 molecules-24-02813-f005:**
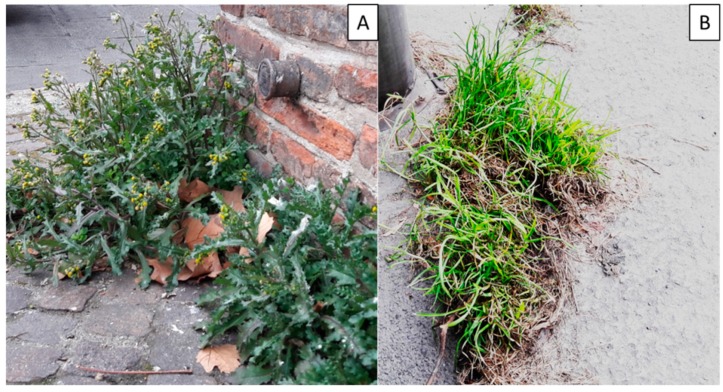
(**A**) *Senecio vulgaris* growing at a busy street crossing in Bologna. (**B**) *Poa annua* growing on the sidewalk in Milan.

**Table 1 molecules-24-02813-t001:** Instrument settings for AAS determination.

Element	Wavelength (nm)	Slit (nm)	Drying Temperature (°C)	Pyrolisis Temperature (°C)	Atomization Temperature (°C)
**Zn (II)**	213.9	0.70	110	700	1800
**Cu (II)**	324.8	0.80	110	1000	2300
**Pb (II)**	283.3	1.05	110	950	1800
**Cr (VI)**	357.9	0.80	110	1650	2500
**Cd (II)**	228.8	1.35	110	850	1650
**Ni (II)**	232.0	1.35	110	1400	2500

**Table 2 molecules-24-02813-t002:** Empirical rules adopted in the evaluation of correlations. Colors indicate the “goodness” of correlation: yellow = significant; orange = relevant; green = high; blue = excellent.

0.3<correlation<0.5	significant
0.5<correlation<0.7	relevant
0.7<correlation<0.9	high
0.9<correlation<1	excellent

**Table 3 molecules-24-02813-t003:** Total and bioavailable concentrations of six heavy metals in the analyzed soils.

Soil	pH	OM(%)	Zn (ppm)		Cu (ppm)		Pb (ppm)		Cr (ppm)		Cd (ppm)		Ni (ppm)	
			Total	Bioavail.	Total	Bioavail.	Total	Bioavail.	Total	Bioavail.	Total	Bioavail.	Total	Bioavail.
**MI3**	7.73	13.1	1200 ± 300	N.A.	390 ± 30	36 ± 3	530 ± 50	28 ± 2	229 ± 10	0.48 ± 0.03	0.48 ± 0.01	0.03 ± 0.01	175 ± 5	3.80 ± 0.02
**MI4**	7.75	9.40	1200 ± 200	N.A.	540 ± 40	113 ± 7	135 ± 2	5.5 ± 0.3	75 ± 1	0.61 ± 0.06	0.39 ± 0.01	0.08 ± 0.01	70 ± 10	3.5 ± 0.1
**MI9**	8.81	9.48	270 ± 40	N.A.	60 ± 10	N.A.	22 ± 3	N.A.	100 ± 70	N.A.	0.40 ± 0.2	N.A.	60 ± 30	N.A.
**BO7**	9.04	7.90	410 ± 70	N.A.	133 ± 7	11 ± 1	110 ± 20	4.3 ± 0.4	40 ± 10	0.15 ± 0.01	0.29 ± 0.01	0.08 ± 0.01	39 ± 4	1.5 ± 0.1
**BO8**	8.91	6.59	510 ± 40	N.A.	110 ± 10	2.6 ± 0.2	120 ± 20	4.6 ± 0.4	150 ± 60	0.19 ± 0.01	0.6 ± 0.1	0.06 ± 0.01	130 ± 90	1.4 ± 0.1
**NAT1**	7.42	11.8	56 ± 3	N.A.	12 ± 2	2.6 ± 0.2	12 ± 4	2.8 ± 0.1	19 ± 1	0.83 ± 0.03	0.10 ± 0.05	0.19 ± 0.02	11 ± 3	0.9 ± 0.1
**NAT5**	8.79	4.57	184 ± 6	N.A.	50 ± 5	3.9 ± 0.4	19 ± 5	0.53 ± 0.01	22 ± 5	0.07 ± 0.01	0.33 ± 0.04	0.04 ± 0.01	49 ± 4	4.3 ± 0.3
**NAT8**	8.79	2.70	110 ± 10	N.A.	19 ± 2	N.A.	8 ± 4	N.A.	570 ± 50	N.A.	0.38 ± 0.03	N.A.	1900 ± 300	N.A.

**Table 4 molecules-24-02813-t004:** Correlation table between BAF_TM and BAF_BM. Colors indicate the “goodness” of correlation: yellow = significant; orange = relevant; green = high; blue = excellent. Information about Zn was not available.

Correlation BAF_TM/BAF_BM	Zn	Cu	Pb	Cr	Cd	Ni
*P. annua*	N.A.	0.33	0.79	−0.03	0.10	0.87
*P. aviculare*	N.A.	0.00	0.78	0.94	0.83	0.81
*S. vulgaris*	N.A.	0.97	1.00	−0.22	0.98	0.97

**Table 5 molecules-24-02813-t005:** Metal concentrations and bioaccumulation factor (BAF) for the three studied species.

Species	Soil	Zn (ppm)				Cu (ppm)				Pb (ppm)				Cr (ppm)				Cd (ppm)				Ni (ppm)			
		Plant			BAF	Plant			BAF	Plant			BAF	Plant			BAF	Plant			BAF	Plant			BAF
*S. vulgaris*	Mi3	17.9	±	0.7	0.01	0.78	±	0.05	0.02	<LoD			N.D.	<LoD			N.D.	0.05	±	0.01	0.02	0.39	±	0.03	0.02
*S. vulgaris*	Mi4	471	±	40	0.32	9.8	±	0.9	0.39	<LoD			N.D.	0.16	±	0.01	0.02	0.05	±	0.01	0.02	0.67	±	0.03	0.03
*S. vulgaris*	Mi9	70	±	1	0.39	8.6	±	0.5	0.28	0.36	±	0.02	0.09	0.51	±	0.01	0.16	0.21	±	0.02	0.01	0.64	±	0.03	0.02
*S. vulgaris*	Bo7	99	±	4	0.17	7.3	±	0.5	0.07	0.77	±	0.02	0.01	0.49	±	0.02	0.02	0.21	±	0.01	2.49	1.25	±	0.01	0.04
*S. vulgaris*	Bo8	5.4	±	0.5	0.01	0.08	±	0.01	0.01	<LoD			N.D.	0.02	±	0.01	0.02	0.06	±	0.01	0.09	0.25	±	0.02	0.02
*S. vulgaris*	Nat8	17	±	1	0.16	2.9	±	0.2	0.16	0.48	±	0.03	0.03	1.14	±	0.07	0.03	0.73	±	0.02	1.86	5.1	±	0.3	0.02
*S. vulgaris*	Nat1	54	±	4	1.02	8.26	±	0.03	0.69	0.42	±	0.03	0.02	0.7	±	0.01	0.05	0.76	±	0.03	0.01	3.07	±	0.06	0.06
*S. vulgaris*	Nat5	22.7	±	0.6	0.13	5.7	±	0.2	0.14	4.96	±	0.03	0.31	0.16	±	0.01	0.01	14.6	±	0.4	0.02	2.4	±	0.1	0.06
*P. aviculare*	Mi3	47	±	3	0.03	14	±	1	0.04	0.82	±	0.03	0.02	1.7	±	0.1	0.01	0.17	±	0.01	0.36	1.93	±	0.08	0.01
*P. aviculare*	Mi4	56	±	2	0.04	13.7	±	0.7	0.03	1.03	±	0.08	0.01	1.58	±	0.08	0.02	0.17	±	0.01	0.44	0.62	±	0.01	0.01
*P. aviculare*	Mi9	30	±	3	0.16	21	±	1	0.22	0.63	±	0.05	0.02	3.52	±	0.07	0.13	0.24	±	0.01	1.86	1.8	±	0.2	0.15
*P. aviculare*	Bo7	57	±	3	0.13	19.8	±	0.2	0.15	0.81	±	0.01	0.01	2.5	±	0.2	0.12	0.47	±	0.04	5.48	0.65	±	0.03	0.02
*P. aviculare*	Bo8	46	±	5	0.20	7.8	±	0.7	0.14	1.02	±	0.08	0.01	1.3	±	0.1	0.08	4.04	±	0.07	1.72	1.23	±	0.05	0.02
*P. aviculare*	Nat8	32	±	2	0.31	39	±	1	2.17	0.12	±	0.01	0.01	0.22	±	0.03	0.02	0.16	±	0.01	0.42	2.2	±	0.2	0.02
*P. aviculare*	Nat1	40	±	2	0.75	3.6	±	0.1	0.30	0.11	±	0.02	N.D.	0.17	±	0.01	0.01	0.35	±	0.01	5.01	1.2	±	0.1	0.06
*P. aviculare*	Nat5	27.4	±	0.6	0.29	3.15	±	0.09	0.14	0.13	±	0.01	N.D.	0.12	±	0.01	0.01	0.33	±	0.02	1.43	1.6	±	0.2	0.03
*P. annua*	Mi3	220	±	10	0.15	1.80	±	0.01	0.01	<LoD			N.D.	<LoD			N.D.	0.35	±	0.01	0.71	4.1	±	0.3	0.03
*P. annua*	Mi4	108	±	8	0.20	14.0	±	0.6	0.07	0.08	±	0.01	0.01	2.33	±	0.01	0.07	0.46	±	0.04	1.92	5.8	±	0.4	0.07
*P. annua*	Mi9	84.0	±	0.4	0.47	14.2	±	0.1	0.15	0.54	±	0.02	0.02	4.14	±	0.05	0.14	0.42	±	0.02	3.31	6.8	±	0.6	0.20
*P. annua*	Bo7	29	±	2	0.10	11.0	±	0.8	0.08	0.17	±	0.03	0.01	0.45	±	0.02	0.01	0.13	±	0.01	0.48	0.82	±	0.01	0.06
*P. annua*	Bo8	119	±	9	0.26	20.2	±	0.3	0.19	0.25	±	0.02	0.01	1.52	±	0.06	0.07	0.23	±	0.02	0.48	2.00	±	0.07	0.14
*P. annua*	Nat8	4.0	±	0.4	0.04	4.1	±	0.4	0.23	0.54	±	0.02	0.03	0.38	±	0.01	0.01	1.53	±	0.01	3.89	17.2	±	0.2	0.02
*P. annua*	Nat1	98	±	5	1.85	5.2	±	0.3	0.43	1.7	±	0.1	0.08	3.1	±	0.1	0.21	0.61	±	0.06	8.57	3.5	±	0.3	0.19
*P. annua*	Nat5	34	±	1	0.37	7.7	±	0.7	0.33	0.41	±	0.03	0.01	1.6	±	0.1	0.06	0.23	±	0.01	1.01	6.27	±	0.03	0.17

**Table 6 molecules-24-02813-t006:** Correlation table between PM and TM. Colours indicate the “goodness” of correlation: yellow = significant; orange = relevant; green = high; blue = excellent.

Correlation TM/PM	Zn	Cu	Pb	Cr	Cd	Ni
*P. annua*	0.64	0.15	−0.47	−0.47	−0.05	0.87
*P. aviculare*	0.59	−0.05	0.50	−0.27	0.61	0.62
*S. vulgaris*	0.56	0.04	−0.28	0.54	−0.02	0.73

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
