# Peer review of "Heavy Metals Bioindication Potential of the Common Weeds *Senecio vulgaris* L., *Polygonum aviculare* L. and *Poa annua* L."

_molecules, 2019, doi:10.3390/molecules24152813_

Round 1
Reviewer 1 Report
This study analyzed the ruderal species Senecio vulgaris L., Polygonum aviculare L. and Poa annua L. to biomonitor Cu, Zn, Cd, Cr, Ni and Pb in multiple environments. It demonstrated the feasibility to use P. annua and S. vulgaris as bioindicators of Ni pollution in soil. The two species were reliable indicators both of total and bioavailable Ni fraction. Although this topic is interesting, the organization and written can be great improved to clarify its novelty and to support its conclusion. Based on its potential interest of readership in Molecules, the referee suggested the major revision before further consideration.
Major:
1 The novelty should be detailed explain in the introduction.
2 Line 48, 49, “The ability of plants to accumulate 48 HMs into their tissues may therefore be used to monitor soil pollution”, Is the enrichment of heavy metals in plants necessarily proportional to the concentration in the soil? The enrichment of heavy metals in plant organs may also be affected by climate, precipitation, pests and diseases?
3 Line 264, 265. What does “MI” and “NAT” mean? It is not stated in the text and in the table. Please write the full name in the footnote of the table. And the Table 3 should use a three-line table.
Minor
1 Please use the "Times New Roman" font in all manuscripts, tables and numbers.
2 Please write the full name in TITLE and at its first appearance, such as Line 3, 4 and 14 “Senecio vulgaris L., Polygonum aviculare L., Poa annua L.”; Line 17 “OM”.
3 Line 24, 25, 308, 327, 334, 351, 391, Change “R” to “R2 or R-square”.
4 Line 134, 140, change “10-mL” to “10 mL”. Line 139, change “2-hours” to “2 hours”.
5 Line 155, 156, in Table 1, the table should use a three-line table, and the right side is incomplete, please complete.
6 Line 236, Table 2 lacks the header line, completes the table, and uses a three-line table.Line 271 “(~10-50 ppm).”, line 272 “~ 100-500 ppm”, line 273 “~ 10-20 ppm”, line 277 “(Ni ~1800 ppm, Cr ~ 500 ppm).” change "~" to "about".
7 Line 304, 305, the table 6 should use a three-line table.
8 Line 416, 417, remove the extra images of the other cars in the upper left corner of Figure A.
9 Line 477, 512, 514, 522, 523, 535, please provide complete, correctly formatted citation in the reference list.
Author Response
We thank the Reviewer for the relevant and constructive comments, which helped us to improve the quality of our work.
We accepted all the comments, and we corrected the manuscript accordingly.
The attached file reports our detailed answers.

Reviewer 2 Report
The study entitled "Heavy Metals Bioindication Potential of the Common Weeds Senecio vulgaris L., Polygonum aviculare L. and Poa annua L." tested three mentioned ruderal species as possible candidates for biomonitoring of Cu, Zn, Cd, Cr, Ni and Pb in 3 different environments (urban, woodland, ultramafic). The manuscript is well organized and relatively well written but some parts should be improved or more elucidated (see below). Overall graphical presentation should be better. Images have low resolution - vector graphics should be used (except of photos).
Similar studies are important from an environmental point of view, but may also help, for example, for remediation of contaminated soils. The submitted work may be a preliminary study, but due to the relatively small amount of data, its contribution is relatively small. The study deserves praise for using statistical tools for data processing. On the contrary, the relatively weak point of the study is the lack of a deep chemical insight, the total concentrations of some elements can be very misleading information. There is insufficient discussion about the existence of different oxidation states of some elements, especially for chromium (Cr(III) / Cr(VI)), but the possible presence of two different forms of lead (Pb(II) and Pb(IV)) is also important. In some cases, the toxicity of inorganic and organometallic forms is fundamentally different, for example lead (tetraethyl lead). It can be understood that these problems go beyond the intended scope of the study and therefore the data was not collected. However, it is not possible to work without this data for a comprehensive view of the matter and a proper understanding of environmental contamination. If the authors did not want to deal with the issue in such detail, they should at least justify their position and admit that it is not completely comprehensive.
Table 1 shows the determination of selected elements including Cr(VI), which is the determination of the element only after digestion (oxidation to the oxidation stage VI). This does not necessarily mean that Cr(VI) in this form is also present in the soil or in plants. The ability to perform speciation in AAS analyzes is limited, but not impossible. While Cr(III) does not pose a major risk, Cr(VI) presents a very high risk. The authors should consider whether it would be appropriate to at least marginally deal with this fact and at least explain why they do not acquire it. Although the article is more environmentally oriented, the chemistry of the process taking place in the soil and plants should not be hidden.
Specific comments:
line 11: HM is not defined (some abbreviation are defined in text, some not - why?; list of abbreviations is placed at line 452).
line 17: OM is not defined (introduced at line 187).
line 24: R = 0.78 is not STRONG linear correlation.
line 59: "iii)" should be in italic
line 98: Typograhical error - "50° C" should be written as "50 °C".
line 101: It is recommended to increase Fig. 1 resolution, it is too low.
line 114: Please see the comment for line 98.
line 117: Properties and source of MilliQ water is not mentioned.
line 118: It shoul be better to write "iron(II)" instead of "iron (II)". Style should be uniform - compare with Table 1.
line 124: Please see the comment for line 98.
line 136: Please see the comment for line 98.
line 137: Please see the comment for line 98.
line 149: Please see the comment for line 98.
line 151: Please see the comment for line 98.
line 155: Table 1. is wrongly cropped
line 155: Please check all values in the Table 1. - number of decimal digits in one collumn should be constant or the number of significant digits should be the same.
line 158: Company, town, country is missing for AAS machine.
line 159: Company, town, country is missing for autosampler.
line 160: Company, town, country is missing for graphite furnace.
line 165: Specific conditions are missing for zinc (for other elements are in Table 1).
line 174: Please specify LOD computation method used in your case.
line 164, 165: In your manuscript you use shlash in units, e.g. "mg/L", more often. Here "L min-1" is used. It is recommended to replace it with "L/min". Moreover correct "minus sign" symbol is not used and unwanted line wrap between "-" and "1" therefore appeared.
line 190: Hard space should be used between number and unit to overcome line wrap between number and unit.
line 197 and 202: "g kg-1" - please see the comment for line 164.
line 198, 199: Please see the comment for line 118.
line 238-245: Some abbreviations (TM, BM, PM and BAF) are used but not all are introduced when used for the first time. E.g. BAF is introduced at line 406. Some are introduced repeatedly. Please check whole document carefully and work with abbreviations according to authors instruction for targed journal.
line 251: It is recommended to increase Fig. 2 resolution, it is too low. Moreover in the case of Fig. 2 the use of vector format will be more convenient.
line 256: "Areas" should be written without capital.
line 267: Again there is a question of bioavailability of different forms (e.g. Cr(III)/Cr(VI)) and its relative concentration (ratio) in total and in bioavailable part.
line 267: If in one collumn can be found two values like "9.4" and "9.48" it is obvious that correct should be "9.40" and "9.48" or "9.4" and "9.5" (see the comment for line 155).
line 288: In Table 4 names should be written in italic.
line 296: In Table 5 names should be written in italic.
line 305: In Table 6 names should be written in italic.
line 316: Please see the comment for line 251.
line 382: "Moreover" instead of "Morover".
line 408: Please unify "et al.". At line 68 is written using italic, at lines 133, 408 and 434 not.
line 416: Please use italic for "Senecio vulgaris" and "Poa annua".
line 442: Please replace "do no correlate" with "do not correlate".
line 604: Something is missing.
Some parts were found to be identical and should be rephrased:
line 30-33, line 40-43, line 48-49, line 157-161, line 167-185 (it is recommended to remove this part and use just additional info and reference), 382-383.
Author Response

(The authors gave the same response as above.)

Round 2
Reviewer 1 Report
It is well revised accordingly. Please use the three-line table for all the tables before publication.